# Transcriptional activities of methanogens and methanotrophs vary with methane emission flux in rice soils under chronic nutrients constraints of phosphorus and potassium

**Rong Sheng[1,3], Anlei Chen[1], Miaomiao Zhang[1], Andrew S Whiteley[2,3], Deepak Kumaresan[2,3], Wenxue Wei[1,3]**

[1]Key laboratory of Agro-ecological Processes in Subtropical Regions and Taoyuan Agro-ecosystem Research Station, Soil Molecular Ecology Section, Institute of Subtropical Agriculture, Chinese Academy of Sciences, Changsha 410125, China

[2]School of Earth and Environment, The University of Western Australia, Perth 6009, Western Australia

[3]ISA-CAS and UWA Joint Laboratory for Soil Systems Biology, Institute of Subtropical Agriculture, Chinese Academy of Sciences, Changsha 410125, China

*Correspondence to*: Wenxue Wei (wenxuewei@isa.ac.cn)

**Abstract** Nutrient status in soil is crucial for the growth and development of plants which indirectly/directly affect the ecophysiological functions of resident soil microorganisms. Soil methanogens and methanotrophs can be affected by soil nutrient availabilities and plant growth, which in turn modulate methane ($CH_4$) emissions. Here, we assessed whether deficits in soil available phosphorus (P) and potassium (K) modulated the activities of methanogens, methanotrophs in a long

term (20 y) experimental system undergoing limitation in either one or both nutrients. Results showed that a large amount of $CH_4$ emitted from paddy soil at rice tillering stage (flooding) while $CH_4$ flux was minimum at ripening stage (drying). Compared to NPK treatment, the soils without P input significantly reduced methane flux rates, whereas without K input did not. Under P limitation, methanotroph transcript copy number significantly increased in tandem with a decrease in methanogen transcript abundance, suggesting that P-deficient induced changes in soil physiochemical properties in tandem with rice plant growth might constrain the activity of methanogens, whereas the methanotrophs might be adaptive to this soil environment. In contrast, lower transcript abundance of both methanogen and methanotrophs were observed in K-deficient soils. Assessments of community structures based upon transcript indicated that soils deficits in P induced greater shifts in the active methanotrophic community than K-deficient soils while similar community structures of active methanogens were observed in both treatments. These results suggested that the population dynamics of methanogens and methanotrophs could vary along with the changes in plant growth states and soil properties induced by nutrient deficiency.

**Keywords**: Methane; Methanogen; Methanotroph; Paddy field

## 1. Introduction

Methane is the second most abundant greenhouse gas, next only to $CO_2$, in the atmosphere and

contributes approximately 20-30% of the global warming effect (IPCC, 2007). The atmospheric concentration of $CH_4$ has increased from a pre-industrial value of 0.715 ppm to 1.824 ppm in 2013 (IPCC, 2007; WMO, 2010), with anthropogenic activities accounting for 60% of the global budget of annual $CH_4$ emission (Insum and Wett, 2008). One significant global source of methane is rice paddy cultivation, covering a worldwide area of 155 million hectares and contributing 10% of the annual anthropogenic $CH_4$ emissions (Nazaries et al., 2013).

The net methane flux is determined by the balance between the activities of methanogens and methanotrophs (Le Mer and Roger, 2001). The biochemistry and molecular biology of both methanogens and methanotrophs has been extensively reviewed (Thauer et al., 2008; Trotsenko and Murrell, 2008). In addition to 16S rRNA gene based assays, functional genes (*mcrA* - encoding the alpha subunit of methyl co-enzyme reductase and the *pmoA* gene, encoding the alpha subunit of membrane bound particulate methane monooxygenase) have been successfully used as genetic markers to target both methanogens and aerobic bacterial methanotrophs (MOB), respectively in the environment (Steinberg and Regan, 2008; McDonald et al., 2008).

In previous DNA-based analyses, it was reported that changes in the population size of methanogens and methanotrophs were closely related to the variations in $CH_4$ production and oxidation potential in paddy soils (Dubey and Singh, 2000; Bao et al., 2014; Mohanty et al., 2014), lake sediments and wetland soils (Liu et al., 2014). However, studies have also indicated that the *mcrA* and *pmoA* gene

copy numbers were not significantly correlated with the activities of $CH_4$ production and oxidation, despite the fact that they responded to environmental disturbance (Ahn et al., 2014; Lee et al., 2014; Zheng et al., 2013). Other investigators have suggested that assessment of methanogen community composition, based on *mcrA* gene diversity, remained highly stable in response to environmental changes, showing no significant correlation with the rate of $CH_4$ production in various soil ecosystems (Ma et al., 2012; Xu et al., 2012; Zhang et al., 2014).

It is reasonable to assume that gene abundance analyses alone may not be adequate to link the methanogenic and/or methanotrophic potential of communities with the methane flux (Zheng et al., 2013; Ma et al., 2012; Yuan et al., 2011). However, the transcriptional analysis (mRNA) of *mcrA* and *pmoA* genes can provide information on the 'active' communities of methanogens and methanotrophs (Ma et al., 2012; Zhang et al., 2014; Freitag and Prosser, 2009; Freitag et al., 2010; Angel et al., 2011). Ma et al. (2012) reported that the abundance of *mcrA* transcripts showed a greater correlation with $CH_4$ production rates compared to the *mcrA* gene copies. Similarly, Ahn et al. (2014) also suggested that the transcript abundance of *mcrA* and *pmoA* genes could correlate with the $CH_4$ emission pattern whilst the gene abundance remained relatively stable in rice paddy soil. Interestingly, a study on peat soils indicated that the abundance of *mcrA* or *pmoA* transcripts alone was not correlated with $CH_4$ flux, instead, the transcript/gene ratios of both *mcrA* and *pmoA* genes actually exhibited a linear correlation with $CH_4$ emission (Freitag et al., 2010).

The nutrient availabilities of nitrogen (N), phosphorus (P) and potassium (K) severely influence soil fertility and crop production (Ogden et al., 2002; Pujos and Morard, 1997). Although previous studies have focussed on the effect of N on methane flux, in particular on methanotrophic activity and diversity (Bodelier et al., 2000), little is known on the effects of both P and K on both methanogens and methanotrophs (extensively reviewed in (Veraart et al., 2015)). Specifically, the P and K deficient agricultural land is about 51% and 12% of the total cultivation area in China, respectively. Previous studies indicated that $CH_4$ emissions in P and K deficient plots are significantly lower than balanced inorganic fertilization plots (Yang et al., 2010; Shang et al., 2011). The long-term paddy rice fertilization field experiment utilized in this study was established in 1990, the rice plants in the plots without P input showed severe P deficiency symptoms and loss of about 55% of yield, and the plants in the plots without K input exhibited clear K deficient symptoms and lost about 20% of yield (Zhao et al., 2011, Shang et al., 2011). However, it is unknown that how the functional microorganisms such as methanogens and methanotrophs respond to the soil P and K exhausting environments.

In this study, we hypothesised that the depleting soil available P and K obviously restricted rice plant growth, and simultaneously, it may also affect the community compositions and functions of methanogens and methanotrophs. Therefore, the long-term paddy rice fertilization field experiment was employed, and the soil and gas samples were collected twice at rice tillering and ripening stage. We subsequently used this multi-level approach to resolve the impact of phosphorus and potassium upon

the community composition and abundance of both resident (DNA based) and active (mRNA-based) methanogens and methanotrophs and its subsequent influence upon overall methane flux.

## 2. Materials and Methods

### 2.1. Experimental site

The experimental site is located within the Taoyuan Agro-ecosystem Research Station of the Chinese Academy of Sciences (28$^°$55′ N, 111$^°$26′ E), Hunan province, China. The area is characterized by a subtropical monsoon climate with an annual average air temperature of 16.5℃ and a mean annual precipitation of 1448 mm. Soil samples were collected from a long-term paddy rice field fertilization experiment established in 1990 (Yang et al., 2010; Chen et al., 2010). The paddy soil was derived from quaternary red clay and the cropping regime was a double rice cropping system. The experiment contained ten treatments with three replicates, organised by randomized blocking design, with each plot of 33 m$^2$. The four treatments selected for this study were as follows: NPK (amended with nitrogen, phosphorus and potassium fertilizers), NK (–P, amended with nitrogen and potassium fertilizers), NP (–K, amended with nitrogen and phosphorus fertilizers) and N (–PK, only amended with nitrogen fertilizer). The annual fertilizers input were urea, superphosphate, and potassium chloride at 182.3 kg N ha$^{-1}$ year$^{-1}$, 39.3 kg P ha$^{-1}$ year$^{-1}$, 197.2 kg K ha$^{-1}$ year$^{-1}$, respectively. For the late rice-cropping season when we sampling, urea was applied with three splits, 40% as basal fertilizer, 50% as tillering fertilizer

and 10% as panicle fertilizer. The P and K fertilizers were applied as basal fertilizers before rice

transplanting. The basal fertilizers were well incorporated into the soil by plowing to 10-20 cm depth 2

days before rice planting, and the top-dressing was surface broadcasted. Consistent with the water

management in local late rice-cropping system, flooding was initiated after early rice harvest before late

rice transplanting, and maintained until 10 days before rice harvesting. During this period, a 7 days

drainage episode was implemented at late tillering stage.

## 2.3. Methane emission measurement and soil sampling

In situ methane fluxes from the experimental field plots were sampled using static chambers

(Shang et al., 2011) at rice tillering (flooding) and ripening stages (drying) during the late rice-cropping

season. The sampling chamber was made of PVC with a size of 60×70×90 cm, which was equipped

with one circulating fan inside to ensure sufficient gas mixing and wrapped with a layer of sponge to

minimize air temperature changes inside the chamber during the period of sampling. After rice

transplant, a PVC frame was fixed into a random site in each plot. The top edge of the frame had a

groove for filling with water to seal the rim of the chamber. Each frame enclosed 6 rice plants and the

height aboveground of the frame is only 5 cm to avoid affecting the growth of rice plants. Gas samples

were taken from the chamber headspace with a 30 mL syringe and stored in pre-evacuated vials

(Labcolimited high Wycombe UK). At each sampling stage, $CH_4$ fluxes were measured in triplicate

plots for all treatments once a day for 3 days. Confirmation of a similar variation trend of $CH_4$ fluxes among treatments was observed during these 3 days, we only presented the data from the third day when soil samples were collected in this study.

In order to further explain the dynamic changes of methane flux in the field, fresh soil samples were collected from the plots immediately after in situ $CH_4$ flux sampling. Five soil columns (0–20 cm depth and 5 cm diameter) were randomly taken from each plot and homogenised. The samples were divided into two aliquots, one immediately frozen in liquid nitrogen and stored at -80 $^o$C for nucleic acid extraction and the remainder were used to analyse soil properties and conduct incubation

experiment to determine methane emission rates under controlled environment. The incubation was carried out as follows: after 24 h pre-incubation at 30 $^o$C, equal amounts of fresh soil samples from each treatment (three replicates) were homogenised and 30 g soil (dry weight) was placed into a 250 mL plastic box that can be sealed. For tillering stage samples, soil water content was adjusted to field flooding condition by maintaining 2 cm free surface water. For the ripening stage samples, water

content in the soils was adjusted to the same level (50% moisture content, w/w) according to the highest water content of the fresh soil samples. Afterwards, the plastic boxes were sealed and incubated at 30 $^o$C. Headspace gas sampling was conducted at 0 and 60 min, respectively, using a 5 mL syringe and stored in pre-evacuated vials (Labcolimited high Wycombe UK). The sampled $CH_4$ was analysed using a gas chromatograph equipped with a FID detector (Agilent 7890A, USA).

## 2.2. Measurement of plant biomass and soil properties

Immediately after gas and soil sampling, six randomly chosen rice plants were harvested. After washing off adhering soil from roots, the plant samples were oven dried to constant weight at 60 $^o$C and aboveground and underground biomasses were estimated separately. Data are standardized to 1 m$^2$ plots. Soil organic carbon (SOC) was determined by $K_2Cr_2O_7$ oxidation (Kalembas and Jenkinson, 1973). Total nitrogen (TN) was measured with Automatic Flow Injection after digestion in $H_2SO_4$. After fusion in NaOH, total phosphorus (TP) and potassium (TK) were measured by Inductively Coupled Plasma Spectrometry (Agilen, USA). After extraction with $NH_4OAc$, available K was determined by Atomic Absorption Spectroscopy (Seal, Germany). Available P (AP) was measured using UV-Vis Spectrophotometer (PerkinElmer, USA) following extraction with 0.5 M $NaHCO_3$. Soil pH was determined at a soil to water ratio of 1: 2.5 (Bao, 2000).

## 2.4. Soil microbial DNA and mRNA extractions

Soil microbial DNA was extracted according to Chen et al (2010) with slight modifications. Briefly, after the addition of lysing solution, MP FastPrep-24 (MP Biomedicals, USA) was used instead of a vortex followed by a 15 min water bath treatment at 68 ℃. DNA concentration and quality were measured using a NanoDrop NA-1000 spectrophotometer (Thermo Scientific, Wilmington, DE, USA).

Extraction of total RNA from soils was performed according to the method described by Mettel et al. (2010). The extracted nucleic acid was rendered DNA free by DNase (Promega, USA) digestion according to the manufacturer's instructions. To remove humic acids, the total RNA was reversibly bound to Q-Sepharose and followed by stepwise elution using 1.5 M NaCl, precipitated with isopropanol and resuspended in TE buffer (pH 8.0). In order to remove the 5S rRNA and remaining salts, an RNeasy MinElute Kit (Qiagen, Germany) was used to further purify the total RNA and mRNA associated only with prokaryotes was captured using, the mRNA-ONLY prokaryotic mRNA isolation kit (Epicentre Biotechnologies, United States). Finally, the enriched 700 ng mRNA was reverse-transcribed to cDNA using the Fermentas K1622 RevertAid™ First Strand cDNA Synthesis Kit (Fermentas, USA) and the resulting cDNA was stored at -80$^{\circ}$C.

## 2.5. Composition and abundance of soil methane-cycling communities

For T-RFLP fingerprinting, primers mals/mcrA-rev (Steinberg and Regan, 2008) and A189F/Mb661R (Holmes et al., 1995) were used for PCR amplification of the *mcrA* and *pmoA* gene, respectively. The PCR reaction solution (50 μL) consisted of 60 ng of DNA template, 0.3 μM of each primer and 25 μL 2 × Power Taq Master Mix (TIANGEN, China). Reaction conditions for the *mcrA* gene included an initial denaturation step at 95 ℃ for 3 min, followed by five cycles of denaturation at 95 ℃ for 30 s, annealing at 48 ℃ for 45 s, and extension at 72 ℃ for 30 s, with a ramp rate of 0.1 ℃ s$^{-1}$

from the annealing to the extension temperature. These initial five cycles were followed with 30 cycles of denaturation at 95 ℃ for 30 s, annealing at 55 ℃ for 45 s, and extension at 72 ℃ for 30 s, followed by a final extension step at 72 ℃ for 10 min. The PCR conditions for *pmoA* gene amplification was as follows: after an initial denaturation step at 95℃ for 5 min, followed by 5 cycles of denaturation at 95℃ for 25 s, annealing at 65℃ for 30 s, extension at 72℃ for 30 s. These initial five cycles were followed with 30 cycles of denaturation at 95℃ for 25 s, annealing at 55℃ for 30 s, and extension at 72℃ for 30 s, followed by a final extension step at 72℃ for 10 min.

T-RFLP analysis was performed at Sangni Corporation (Shanghai, China) using an ABI Prism 3100 Genetic Analyzer. T-RFLP profiles for *mcrA* and *pmoA* genes were generated with the endonucleases *HaeIII* (Fermentas, USA) and *HhaI* (Fermentas, USA), respectively. Data analysis of the resultant T-RFLP profiles was performed using PeakScan (version 1.0, Applied Biosystems, Inc.). Fragments with a signal above 1% of the sum of all peak heights were included and peak positions that differed in size by $\leq$ 2 bp in an individual profile were binned and considered as one fragment. Minimum T-RF size for inclusion within the cluster analysis was set at 50 bp or larger.

Local databases of *mcrA* and *pmoA* gene sequences were constructed using over 2,000 downloaded from Functional Gene Pipeline/Repository (FGPR, http://fungene.cme.msu.edu/) and National Center for Biotechnology Information (NCBI, http://www.ncbi.nlm.nih.gov/). *In silico* digestion was performed on these sequences using restriction endonuclease sequences and the T-RFs were assigned to

specific methanogenic and MOB lineages, which was subsequently used to predict and verify the

assignment of individual T-RFs in this study.

For real-time quantitative PCR, *mcrA* and *pmoA* qPCR were performed using the mals/mcrA-rev

(Steinberg and Regan, 2008) and A189F/Mb661R (Kolb et al., 2003) primer pairs, respectively.

Real-time PCR assays were performed in a volume of 10 μL containing 5 μL 2 × SYBR Premix Ex Taq

TM (Takara, Japan), 150 nmol L$^{-1}$ forward and reverse primers and 5 ng of template DNA. Thermal

cycling conditions for the two genes were also the same as described for the T-RFLP analysis. The

standard curves for the *mcrA* and *pmoA* genes were created using a 10–fold dilution series of plasmids

containing the target gene of interest derived by PCR and cloning from soil.

For determination of absolute quantities of *mcrA* and *pmoA* transcripts, the quantitative PCR was

performed using 10 ng of cDNA template. The standard was prepared from *in vitro* transcription of

*mcrA* and *pmoA* clones derived from soil using the Riboprobe *in vitro* Transcription System (Promega)

according to the manufacturer's instructions. The *in vitro* transcript was purified by phenol-chloroform

extraction and quantified using a RiboGreen RNA quantification kit (Invitrogen). The resultant

transcripts were reverse transcribed as described above and a dilution series (10 fold) of cDNA was

used as the standard.


**2.6. Statistical analysis**

Soil properties such as pH, soil organic carbon and total nitrogen together with gene abundance between the treatments were compared by ANOVA analysis using the Statistical Package for Social Science (SPSS 13, SPSS Inc., Chicago, IL, USA). Significance among means was identified using least significant differences. Pearson correlation analysis between $CH_4$ flux, soil properties, plant biomass and population size of resident and active methanogens and methanotrophs was also performed using SPSS. Redundancy analysis (RDA) was used to characterize the relationship between soil properties, plant biomass and the community structures of methanogens and methanotrophs using CANOCO statistical package for Windows 4.5 (Biometris, Wageningen, Netherlands). A Mantel test based on 499 random permutations was used to examine the significant correlations between the differences in soil properties plant biomass and microbial communities.

## 3. Results

### 3.1. Influence of P and K deficiencies on soil properties and plant biomass

Compared to the NPK treatment, –P and –PK treatments induced significant decreases in soil organic carbon (SOC), total phosphorus (TP) and available P (AP) content, whereas –K only caused significant decline in available potassium (AK) content (Table 1). Significant lower plant biomass were also observed in –P, –K and –PK plots compared to NPK treatment, suggesting that deficit in soil P and K availability had restricted the growth of rice plant (Fig. 1). Especially, at the rice tillering stage, the –P

treatment revealed a reduction of 41% and 28% ($P < 0.01$) in aboveground and belowground plant

biomass, respectively.

### 3.2. Influence of P and K deficits on $CH_4$ flux

The measurements from the field plots and soil incubation showed that large amount of $CH_4$

emission was detected at tillering stage while it was at very low level at ripening stage (Fig. 1). The

methane emission rates at tillering stage exhibited that the NPK treatment possessed the highest rate

while the lowest appeared in the −P treatment which was significantly different from NPK ($P < 0.05$).

The $CH_4$ flux in −K treatment were not significantly different from NPK ($P > 0.05$). Meanwhile, the −

PK treatment showed significantly higher $CH_4$ flux than −P ($P < 0.05$) but less than NPK treatment.


### 3.3. Shifts of methanogenic populations and transcripts under exhausting soil available P and K circumstance

When assessing the abundance of the *mcrA* gene, based upon both DNA- and mRNA-based

analyses, we observed significantly higher gene copy numbers at the tillering stage when compared to

the ripening stage across all the treatments (Fig. 2a). At the tillering stage, both −P and −K did not

significantly modulate the abundance of *mcrA* gene copy numbers when compared to the NPK

treatment. However, at the transcription level the treatments of −P, −K and −PK revealed significant

decreases in *mcrA* transcript abundance in comparison with NPK ($P < 0.05$, Fig. 2b). Although no significant difference in *mcrA* transcript abundance was observed between –K and –P treatments ($P > 0.05$), the –PK treatment exhibited lower *mcrA* transcript abundance when compared to both –K and –P treatments (Fig. 2b). Similar to the the lower methane emissions observed at the ripening stage, all the treatments revealed lower *mcrA* transcript copy numbers under both –P and –K treatments, suggesting, as above, that the effect of nutrient limitation on the gene expression was independent of the strength of methane flux or rice cultivation stage (Fig. 2b).

The T-RFLP patterns of resident methanogenic community structures at the rice tillering stage were relatively stable in response to both –P and –K deficits (Fig. 3a, Fig. 4a). However, both –P and –K treatments induced shifts in the active community composition of methanogens when assessing community structure at mRNA level (Fig. 3a, Fig. 4a). These two treatments severely limited the expression of *mcrA* from less abundant members of methanogens represented by T-RFs 118, 208, 277 and 292 bp, but stimulated the activities of other methanogens represented by T-RFs 95 and 202 bp (Fig. 3a). The active methanogen community composition of –PK was roughly similar with that in –P and –K plots, besides that, the further shifts also happened in this treatment, such as the methanogens represented by T-RF 216 bp, corresponding to uncultivated archaeal methanogens, was relatively less abundant in –PK when compared to –P and –K treatments. At ripening stage, both DNA- and mRNA-based analyses revealed similar community compositions as that at the tillering stage, and –P

and –K exhibited similar effects on methanogen community compositions (Fig. 3a, Supplementary Fig. 1a).

## 3.4. Shifts of methanotrophic populations and transcripts under depleting soil available P and K circumstance

For the resident (DNA) MOB abundance, no significant differences were detected between treatments at the tillering stage. Similarly, at the ripening stage, except for the –P treatment, all the treatments revealed no significant differences between observed MOB abundance ($P > 0.05$, Fig. 2c). However, the abundance of the active (mRNA) MOB of the treatments without applying P and K showed a different picture (Fig. 2d). At the tillering stage, the –P treatment showed a significant increase (~85%) in *pmoA* transcript abundance, whereas –K led to a significant reduction (~75%) in *pmoA* transcript when compared to the NPK treatment (Fig. 2d). The *pmoA* transcripts in –PK treatment only accounted for 24% of the NPK treatment and was the lowest among all the treatments. At ripening stage, a different trend was observed with potassium deficits resulting in a significant increase in *pmoA* transcript abundance compared to NPK, –P and –PK treatments.

The MOB community compositions, based on both DNA and mRNA analyses, revealed differential responses under depleting soil available P and K conditions (Fig. 3b). At the tillering stage, both –P and –PK treatments displayed similar T-RFLP patterns but different from the MOB community

compositions in –K plots, especially at the mRNA level (Fig. 4b). –P and –PK treatments resulted in community shifts within the active MOB, particularly T-RF 150 bp, predicted to represent the type I methanotroph *Methylococcus/Methylocaldum*. Transcript abundances of this methanotrophs increased 10 folds, whilst transcripts representing likely members of the genus *Methylococcus* (T-RF 108 bp) significantly reduced in –P and –PK treatment compared to NPK ($P < 0.05$). Meanwhile, –P and –PK treatments induced a decline in the relative abundance of an unknown type II methanotroph genus or *Methylosinus/Methylocystis* (T-RF 81 bp). For other taxa, T-RF 143 bp was observed in –P and –PK treatments. The treatment without K input resulted in a substantial reduction of the relative abundance of T-RF 128 bp (corresponding to several genera, including type I and type II methanotrophs) in resident MOB community (Fig. 3b).

At the ripening stage, –P also induced a significant increase in the relative abundance of T-RF 249 bp within the resident MOB community and a significant reduction of T-RFs 70 bp and 108 bp in the active methanotrophs. Remarkably, –K not only caused significant increase in the relative abundance of T-RF 81 bp in the resident MOB community but also increased the relative abundance of the T-RF 70 bp in the active MOB populations (Fig. 3b). In addition, –PK showed similar T-RFLP pattern to –K (Supplementary Fig. 1b).

**3.5. Correlation between methanogenic, methanotrophic populations, soil properties and CH$_4$ flux**

Correlation analysis indicated that the $CH_4$ fluxes from field and soil incubation were significantly correlated to the transcript ratio of *mcrA/pmoA* ($P < 0.05$ and $P < 0.01$, respectively, Table 2) at the tillering stage. In addition, $CH_4$ flux from soil incubation was also significantly correlated with the contents of both total and available phosphorus (TP and AP, $P < 0.01$), SOC ($P < 0.05$) and plant biomass ($P < 0.05$). Redundancy analysis (RDA) indicated that P-deficient induced changes in soil physiochemical properties, such as SOC, TP, AP contents in tandem with plant biomass, were important factors driving community structure shifts of active (mRNA based) methanogens and methanotrophs (Fig. 5).

## 4. Discussion

Phosphorus and potassium availability have been known to influence methane emissions from peat (Aerts and Toet, 1997) and paddy soils (Yang et al., 2010; Shang et al., 2011; Han et al., 2002). In this study, since the treatments of –P, –K and –PK had been continuously grown rice for 20 years without applying P, K and PK fertilizers correspondingly, the soil Olsen-P and available K concentrations have reached stable minimum levels due to the exhausting effect of plants (Shang et al., 2011). The rice plants showed severe P and K deficiency symptoms in the –P and –K plots, respectively, and the yields reduced significantly in the sampling year (Zhao et al., 2011; Shang et al., 2011). We observed that the soils without P input also induced significant reduction in $CH_4$ emission, whereas –K treatment did not

show clear influence on net methane flux when compared to NPK plots. Since methane emission is the

consequence of the activities of both methanogenic and methanotrophic populations, whether these two

functional groups were also inhibited just like the rice plants under such poor soil P and K nutritional

status is unknown. Although the soil available P and K were at depleting levels after 20 years

experiment, the question is why these two element deficits led to different effects on $CH_4$ emission.

Our results indicated no significant correlation between $CH_4$ flux and the abundance of

methanogens and methanotrophs population sizes (based on DNA), similarly, it was reported that the

abundance of the *pmoA* gene was not correlated to soil methane oxidation rates in paddy fields (Zheng

et al 2013). Theses phenomena suggested that the population sizes of both methanogens and

methanotrophs would be relatively stable in relation to the chemical fertilizations. The differences of

$CH_4$ fluxes caused by the treatments would be strongly linked to the behaviours of their active

communities.

The mRNA-based assessments indicated that the abundances of both active methanogens and

methanotrophs, represented by transcript abundance of *mcrA* and *pmoA*, were more significantly

influenced by the fertilization regimes compared to DNA-based approaches at rice tillering stage when

high $CH_4$ emission was observed, and the $CH_4$ flux was closely related with the transcript ratio of

*mcrA*/*pmoA* ($r^2$=0.682; $P < 0.05$). These clearly expressed that the active methanogenic and

methotrophic communities rather than the whole populations were more sensitively responded to the

soil nutrient status. Although other studies also reported that the community structures based on DNA analysis could respond to soil environmental changes and they could reflect the existing state of functional groups (Ahn et al., 2014; Lee et al., 2014; Zheng et al., 2013), the analysis based on gene transcripts are increasingly reported to provide more useful information in understanding the *in situ* activities of functional microbial communities than the DNA analysis, as gene transcripts are indicative for the active groups against a large resident microbial population (Nicolaisen et al., 2008; Nicol et al., 2008; Freitag et al., 2010).

It was determined that although –P and –K treatments resulted in similar reductions in *mcrA* transcript abundance with similar transcript composition at the tillering stage, they induced different consequences for *pmoA*-containing methanotrophic communities at the transcript level. P deficits caused a significant increase in *pmoA* transcript abundance and also influenced the active methanotrophic community structure. On the contrary, K deficits induced significant reduction in *pmoA* transcript abundance but did not affect the community compositions. The distinct responses of active methanogens and methanotrophs to the P and K limitations are likely to be linked to the difference in their adaptation and response strategies. Phosphorus is an essential life element that is a crucial component of nucleotides and energetic material, such as ATP (Rausch and Bucher, 2002). Phosphorus deficiency can affect both plant and microorganisms, but the critical levels might different. Due to the diverse species of each functional community and their differential adaptabilities to low level of soil P

content (Chauhan et al., 2012), the species within the functional group might possess varied strategies. In the present study, the T-RF representing the genus *Methylococcus*/*Methylocaldum* was markedly enriched within the –P treatment (Fig. 3b). Previous studies have reported that *Methylococcus* and *Methylocaldum* sp. were dominant members of MOB communities in low P oligotrophic soil (Chauhan et al., 2012). Although we do not know the real mechanisms about the enrichment of *Methylococcus*/*Methylocaldum* under such a poor soil P nutritional status, it could be speculated that the possible adaptations of these MOB groups to a P deficient environment might be attributed to one or more adaptive strategies. First, the possession of high-affinity P transporters, capable of producing P-liberating enzymes, as has been documented previously (Veraart et al., 2015; Sebastian and Ammerman, 2009). Second, P use minimisation through low P containing membranes using non-phosphorus lipids (Van Mooy et al., 2009) or smaller genomes and lower RNA content, which can minimize their P-requirements may explain their ability to thrive in low P environments (Sterner and Elser, 2002). In contrast, potassium plays important roles in the activities of enzymes and cell osmotic pressure (Page and Cear, 2006), lacking K may influence the activities of the cells and the expression of functional gens. Thus, as a consequence, the copy numbers of *pmoA* transcripts were sharply decreased under K deficits condition but the compositions were not clearly impacted.

In addition, we focused on the analysis of the possible contributions of methanogens and methanotrophs on methane emission in relation to the soil P and K status, but in fact the plant biomass

was also affected. Although we observed that $CH_4$ emission was significantly related to plant biomass,

$CH_4$ emissions did not always rely on the plant biomass. For instance, the crop yield was significantly

different but the $CH_4$ emission was similar between NPK and –PK treatments. Similar result was also

detected by Shang et al. (2011). So, the mitigation of $CH_4$ emission under very low soil P content might

be influenced by both poor P nutrition of methanogens and methanotrophs and low plant biomass.

Besides, soil water management has been widely known to play an important role in regulating

$CH_4$ emission (Cai et al., 1997; Nishimura et al., 2004; Towprayoon et al., 2005). We observed that the

$CH_4$ flux were much lower at rice ripening stage when soil was drying than that at tillering stage when

soil was under flooding. These phenomena were also reported by previous studies that showing

midseason drainage and the disappearance of the water layer induced significant decline in methane

emission flux, which might associated with the reduction in methane production and increase in the

oxidation of $CH_4$ under drying soil environment (Nishimura et al., 2004; Towprayoon et al., 2005).

It should be noted that previous studies have documented that soils derived from different parent

materials may possess different initial microbial community structure (Ulrich and Becker, 2006; Sheng

et al., 2015). Agricultural practices such as crop planting, fertilization and irrigation can modify the

initial microbial community structures to some extent based on the initial microbial communities (Fierer

et al., 2003). Therefore, different soils may have different methanogenic and MOB community

composition structures, their shift patterns in different soils responded to low P availability may vary

among different soil types. The variations of the methanogenic and MOB communities in responding to the depleting P and K levels in the paddy soil derived from quaternary red clay may be transferrable to other soils in tendency, but the varying species might be obviously different.


## 5. Conclusions

P-deficient soils showed significantly lower $CH_4$ flux. This might be attributed to the restriction of methanogens and the stimulation of methanotrophs that could have adapted to changes in soil physiochemical properties in association with rice plant growth under chronic nutrient constraints. In contrast, K-deficient did not affect the $CH_4$ flux, which might be caused by the reductions of both methanogenic and methanotrophic activities. Comparatively, more variations within community composition of the active methanotrophs were observed in P-deficient soils than that in K-deficient soils, whereas both P- and K-deficient soils shared similar active methanogenic community structures. We have observed these effects in our quaternary red soils, but to what extent it can be generalized remains unclear, considering the remarkable soil heterogeneity.



**Acknowledgements**

These efforts were supported by the National Research Foundation of China (grant numbers 41330856, 41501277) and the Chinese Academy of Sciences Strategic Leading Science and Technology

Projects (grant number XDB15020200). We also acknowledge funding from the Dept. of Premier and

Cabinet and the University of Western Australia under the Western Australian Fellowships Program to

ASW and the University of Western Australia to DK within the CAS-UWA Joint Laboratory on Soil

Systems Biology.

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

**Table 1** The basic characteristics of the examined paddy soil under different fertilization treatments at rice

tillering stage and rice ripening stage.

| Treatment[b] | Soil properties[a] | | | | | | |
|---|---|---|---|---|---|---|---|
| | SOC | TN | TP | TK | AP | AK | pH |
| | (g kg$^{-1}$) | (N g kg$^{-1}$) | (P g kg$^{-1}$) | (K g kg$^{-1}$) | (P mg kg$^{-1}$) | (K mg kg$^{-1}$) | (1:2.5 H$_2$O) |
| NPK | 20.53±1.01a[c] | 1.91±0.09ab | 0.69±0.05a | 13.69±0.10a | 12.23±1.51b | 135.40±33.98a | 5.15±0.15a |
| –K | 20.51±0.36a | 2.10±0.16a | 0.66±0.06a | 13.77±0.46a | 14.38±1.56a | 63.40±5.38b | 5.16±0.15a |
| –P | 18.60±0.06b | 1.86±0.03b | 0.42±0.06b | 14.02±0.31a | 4.45±1.03c | 138.83±35.93a | 5.27±0.23a |
| –PK | 17.94±0.11b | 1.96±0.06ab | 0.43±0.06b | 13.76±0.11a | 4.13±0.37c | 59.28±3.61b | 5.19±0.20a |

[a] Soil properties: *SOC,* total carbon, *TN,* total nitrogen, *TK,* total potassium, *TP,* total phosphorus, *AP,* available

phosphorus and *AK,* available potassium.

[b]Treatments: *NPK,* balance chemical fertilization; *–K,* potassium deficient; *–P,* phosphorus deficient; *–PK,*

phosphorus and potassium deficient.

[c] Significant differences ($P < 0.05$) between treatments are shown with letters a, b, or c; mean ± SEM, n=3 for

each treatment.

**Table 2** Correlation between CH$_4$ flux and methanogenic, methanotrophic populations and soil properties.

| | Sampling stage | mcrA/pmoA (gene) | mcrA/pmoA (transcripts) | SOC | TN | TP | AP | Plant biomass aboveground | belowground |
|---|---|---|---|---|---|---|---|---|---|
| CH$_4$ flux (in situ) | Tillering | 0.167 | 0.682 | 0.480 | 0.134 | 0.548 | 0.328 | 0.548 | 0.328 |
| | Ripening | - | - | - | - | - | - | - | - |
| CH$_4$ flux (in lab) | Tillering | 0.559 | 0.833** | 0.620* | 0.576* | 0.739** | 0.794** | 0.739** | 0.794** |
| | Ripening | 0.738** | 0.441 | 0.723** | 0.479 | 0.876** | 0.845** | 0.876** | 0.845** |

\* Correlation is significant at the 0.05 level.

\*\* Correlation is significant at the 0.01 level.

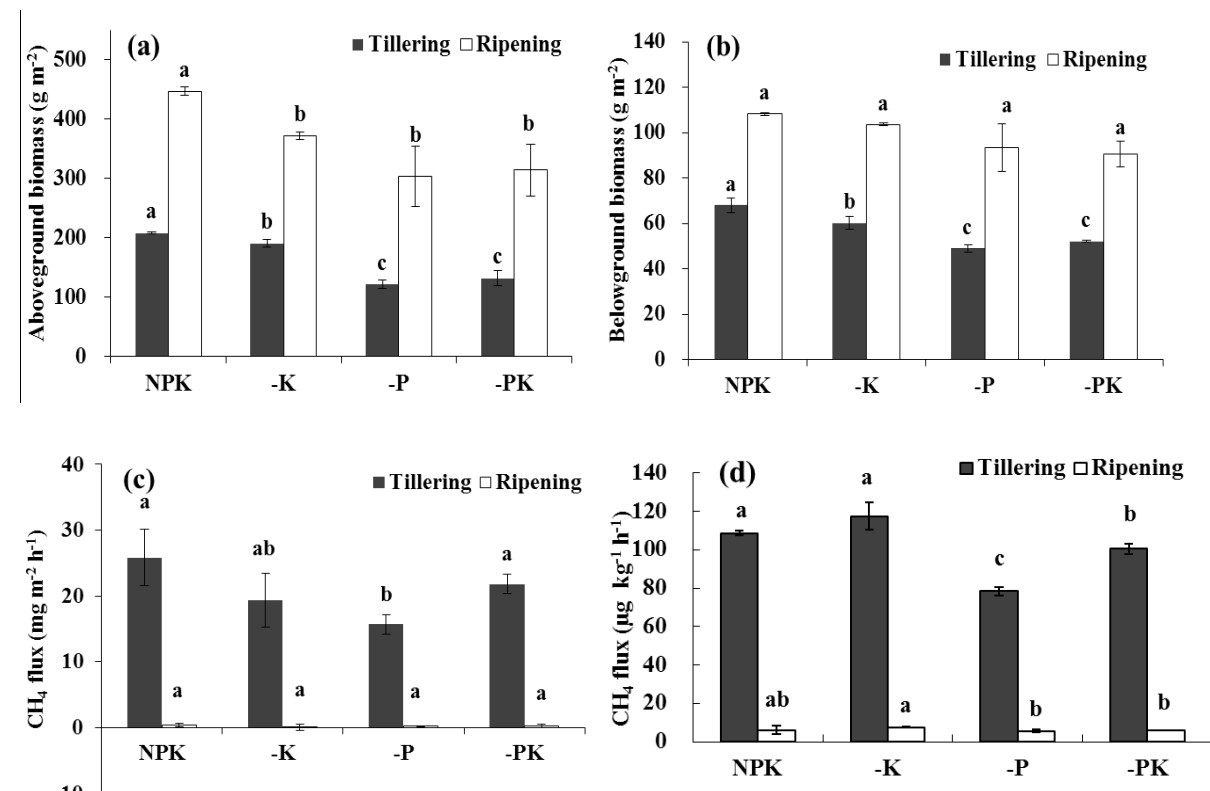


Fig. 1 Aboveground (a) and belowground (b) plant biomass and methane flux from field plots (c) and soil incubation (d) at rice tillering stage and rice ripening stage. *NPK,* balance chemical fertilization; *–K,* potassium deficient; *–P,* phosphorus deficient; *–PK,* phosphorus and potassium deficient. Significant differences ($P < 0.05$) between the treatments are shown with letters a, b, or c; mean $\pm$ SEM, n=3 for each treatment. Statistical analysis for soils from tillering stage and ripening stage were conducted separately.


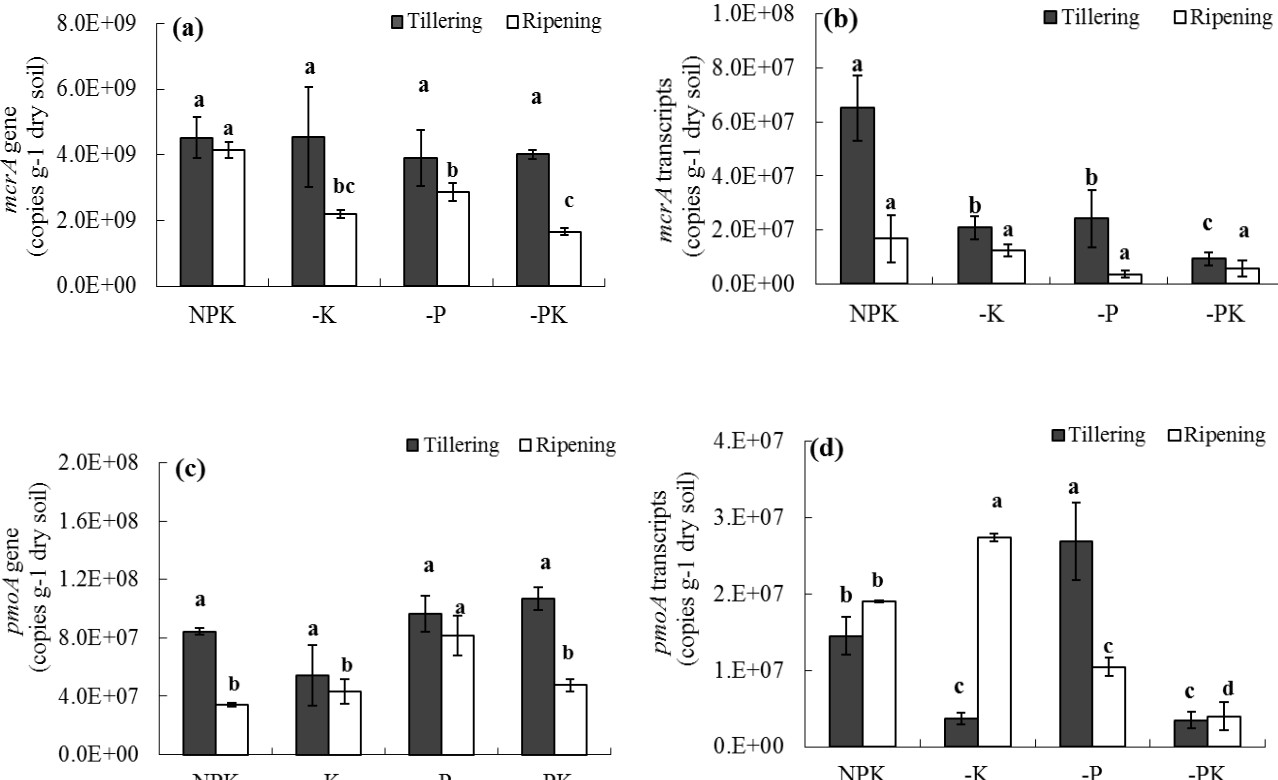


**Fig. 2** Copy numbers of *mcrA* gene (a) and gene transcripts (b) and *pmoA* gene (c) and gene transcripts (d) in relation to nutrient P and K deficient condition. ***NPK***, balance chemical fertilization; ***–K***, potassium deficient; ***–P***, phosphorus deficient; ***–PK***, phosphorus and potassium deficient.

Significant differences ($P < 0.05$) between the soils are shown using letters a, b, or c. Statistical analysis between soils from tillering and ripening stage was performed separately. Soils with the same letter at each depth are not significantly different at the $P < 0.05$ level.

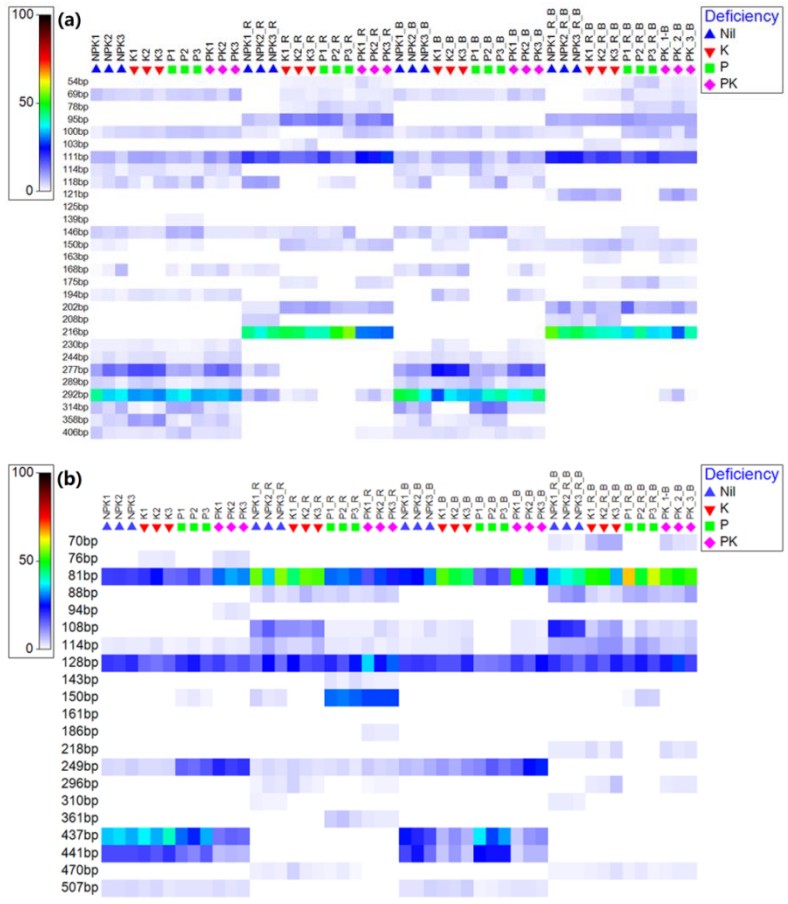


**Fig. 3** Heatmap of *mcrA*-(a) and *pmoA*-based (b) T-RFLP profiles showing average relative abundances

of *mcrA* T-RFs with endonuclease *HaeIII* and *pmoA* T-RFs with endonuclease *HhaI* in soils. The

relative abundance of T-RFs is given as a percentage of the total peak height. Fragment sizes

within the graph indicate the sizes (bp) of the experimental T-RFs by T-RFLP. Letters "R" after the

treatments indicate samples from mRNA-derived profile, and letters "B" indicate the samples from

ripening stage. ***NPK***, balance chemical fertilization; ***K***, potassium deficient; ***P***, phosphorus

deficient; ***PK***, phosphorus and potassium deficient.

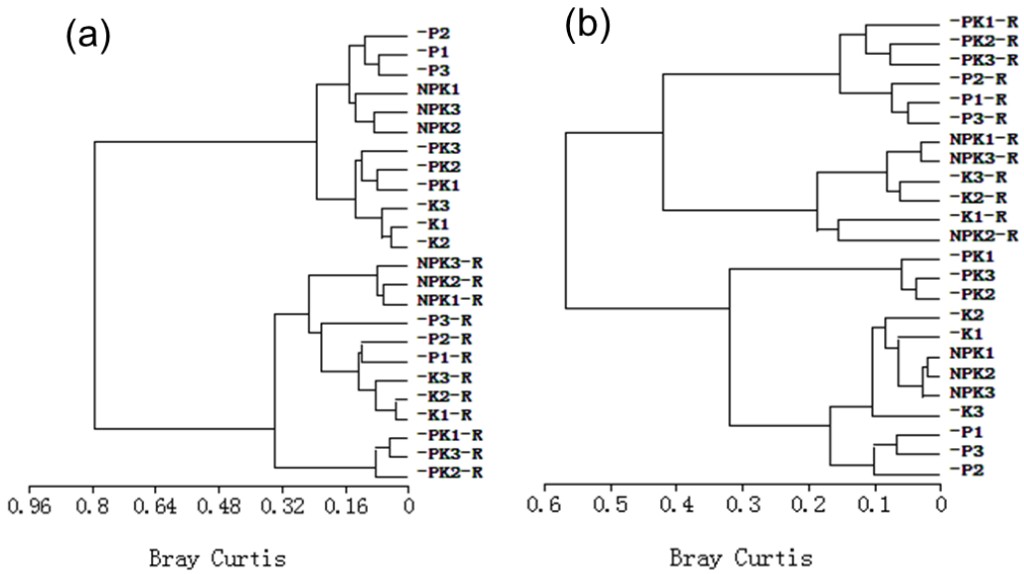

Fig. 4 Cluster analysis of *mcrA*-(a) and *pmoA*-based (b) T-RFLP profiles from rice tillering stage.

Letters "R" after the treatments indicate samples from mRNA-derived profile. ***NPK***, balance chemical fertilization; ***–K***, potassium deficient; ***–P***, phosphorus deficient; ***–PK***, phosphorus and potassium deficient.

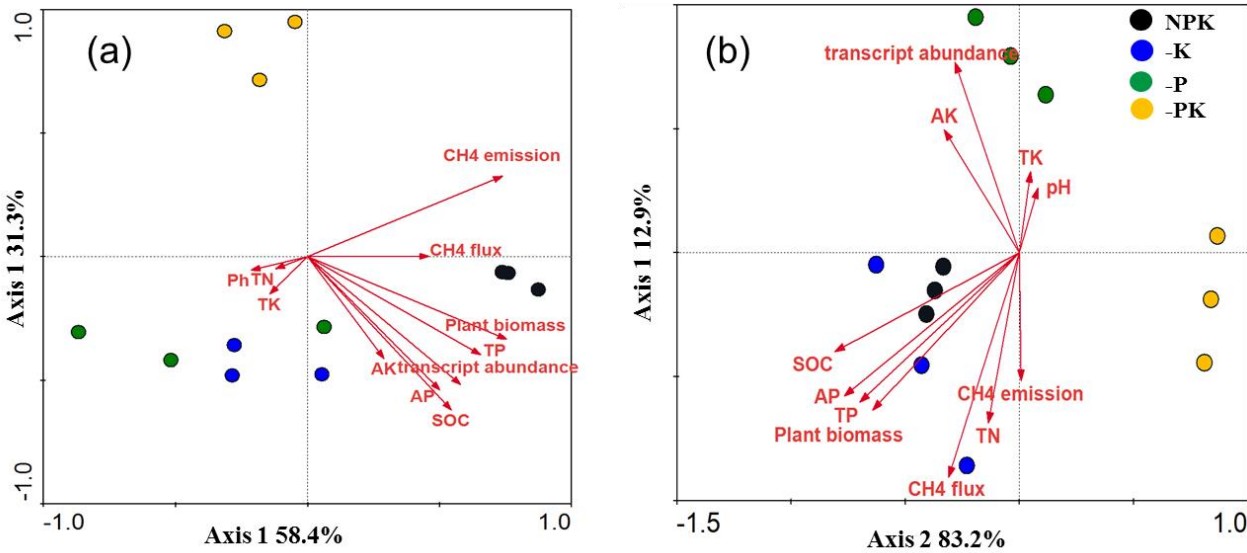

Fig. 5 Redundancy analysis indicating relationships between soil properties, plant biomass and community structures of *mcrA* (a) and *pmoA* (b) gene transcripts from rice tillering stage. ***NPK***, balance chemical fertilization; ***–K***, potassium deficient; ***–P***, phosphorus deficient; ***–PK***, phosphorus and potassium deficient. ***CH₄ emission***, methane flux from field plots; ***CH₄ flux***, methane flux from soil incubation.