# Peer review of "Transcriptional activities of methanogens and methanotrophs vary with methane emission flux in rice soils under chronic nutrients constraints of phosphorus and potassium"

_Biogeosciences, 2016_

## Editor Comment (EC1) · Z Jia (Editor) · 17 Aug 2016

(1) The abstract. please clearly describe the patterns of methane emission fluxes in the field as a starting point in your abstract. And then explain the discrepancy of methane flux in the field by the variations of mcrA and pmoA genes and their transcripts in the context of P and K nutrient status. The soil properties and plant biomass can also be used to interpret the methane flux variations, and the abstract can be concluded with the statistically most significant factors that may determine methane flux in the field. (2) The title. The title might be reasoned whether it can actually reflect the most important findings in this study. Indeed it seems for me that soil water management played a much more important roles than nutrient fertilizers (3) The relevant reference might be added as follows. Veraart, A. J., Steenbergh, A. K., Ho, A., Kim, S. Y., & Bodelier, P. L. E. (2015). Beyond nitrogen: the importance of phosphorus for CH4 oxidation in soils and sediments. Geoderma, 259-260(December), 337-346.doi:10.1016/j.geoderma.2015.03.025

---

## Referee Comment (RC1) · Anonymous Referee #1 · 21 Sep 2016

General comments

The authors performed an interesting study on effects of P and K availability in rice paddy soils on methane emission and methanogen and methanotroph presence and community composition. They find reduced CH4 emission in low P plots, which they attribute to higher methanotroph activity and lower methanogen activity. Effects of potassium on CH4 emission are less pronounced. Copy numbers of mcrA and pmoA genes are not linked to CH4 emission and hardly show a response to fertilization treatments. Transcripts of pmoA showed differences in the active MOB community between treatments, also dependent on the rice growth stage.

Strong points of the manuscript are the inclusion of mcrA and pmoA transcript analysis in combination with CH4-flux data, and the use of long-term fertilization plots. However,

I would like to see improvement on the following issues:

Introduction/Discussion

To my understanding, it is quite clear when P or K are limiting plants, but concentrations at which they become limiting to microbes are far less understood. Therefore, I recommend to use terms like 'P/K deficient' with care. For example, your results point at an increase in MOB activity in low P plots, perhaps indicating not so much P-limitation, but an alleviation of excess P? Also, effects may arise from altered (C):N:P:K stoichiometry, rather than concentrations in itself.

Discuss how, because different MOB respond in different ways, results may strongly depend on the initial community composition. Different soils may react in different ways.

Methods

The CH4-flux method is poorly described. Please provide more detail on the method. Where and how were the samples taken? Did the chambers include rice plants? Did you measure time-series? From how many static chambers per plot? How many replicate gas samples? How many replicates in time? How were total and available P and K determined? Do they reflect availability to plants? / to which extent is P or K unavailable to plants available to microbes?

Results

The figures can be improved. It would be helpful to show hierarchical clustering of the samples based on their T-RFLP profiles, to show which samples/treatments are most similar (per sample class). Add gene names and DNA vs mRNA copies inside the panels or on the y-axes of the graphs.

Show correlations between CH4-flux and DNA and mRNA copies, and also present the relation between CH4 flux and mcrA/pmoA transcripts here. You refer to these in the discussion but they are missing in the results section.

Discussion

Please also better explain why one would expect DNA copy numbers to be less indicative of community functioning than transcripts.

I am missing some discussion on what these results mean in terms of CH4-mitigation potential? Low emissions seem to come at the expense of plant biomass (and possibly nutritional value?).

Specific comments Line 31: I would end this sentence at 'transcriptional level', as the relation between 'population size' and DNA copies is debatable. Line 125, why? Line 254 This seems to conflict with the previous sentence, where members of methylococcus increased. Are these T-RFs representing different species, meaning some methylococcus species increase whereas others decrease? Line 280 describe how they were influenced by the fertilizer regime Line 286 the 'size' of the resident communities is hardly affected. It would be interesting to also discuss the effect of the growth stage of rice on CH4 flux and methanogen and MOB communities. Line 303. How can you be sure that they were P deficient? Line 325: Add that effects are species specific, different soils may show different effects

Technical comments Line 78, round to whole numbers Line 87, key nutrients -> phosphorus and potassium Line 113, after washing off Line 204 different

---

## Author Comment (AC1) · 25 Oct 2016

Dear Editor, Thank you for your comments and suggestions on our manuscript. We have revised the manuscript accordingly, and highlighted the changes in the revised version. The detailed corrections are listed below point by point:

**Comment 1: The abstract. please clearly describe the patterns of methane emission fluxes in the field as a starting point in your abstract. And then explain the discrepancy of methane flux in the field by the variations of mcrA and pmoA genes and their transcripts in the context of P and K nutrient status. The soil properties and plant biomass can also be used to interpret the methane flux variations, and the abstract can be concluded with the statistically most significant factors that may determine methane flux in the field.**

[Figure]

Answer: The abstract was rewrite as follows: Nutrient status in soil is crucial for the growth and development of resident microorganisms. Soil methanogens and methanotrophs can be affected by soil nutrient availabilities, which in turn modulate methane (CH4) emissions. However, it is not clear about the influence of nutrient limitation on the methanogenic and methanotrophic communities and their functions. We assessed whether deficits in soil available phosphorus (P) and potassium (K) modulated the activities of methanogens, methanotrophs in a long term (20 y) experimental system undergoing limitation in either one or both nutrients. Results showed that a large amount of CH4 emitted from paddy soil at rice tillering stage (flooding) while CH4 flux was minimum at ripening stage (drying). Compared to NPK treatment, the soils without P input significantly reduced methane flux rates, whereas without K input did not. Under P limitation, methanotroph transcript copy number significantly increased in tandem with a decrease in methanogen transcript abundance, suggesting that soils lacking P induced CH4 emission reduction would be via reduced methane production in tandem with increased methane consumption potential. In contrast, K deficits reduced both methanogen and methanotrophs transcript abundance. Assessments of community structures based upon transcript indicated the treatment without P amendment induced greater shifts in the active methanotrophic community than for K deficits while similar community structures of active methanogens were observed in both treatments. Correlation analysis indicated that soil phosphorus availability, SOC contents and plant biomass were important factors in regulating CH4 emission from the field.

**Comment 2: The title. The title might be reasoned whether it can actually reflect the most important findings in this study. Indeed it seems for me that soil water management played a much more important roles than nutrient fertilizers**

Answer: Soil water management indeed plays an important role in regulating CH4 emission, and it has been already proved by previous studies (Cai et al., 1997; Nishimura et al., 2004; Towprayoon et al., 2005). In this study, we focused on the nutrient status on the process of CH4 emission at two rice growing stages. The long-term

paddy rice fertilization field experiment utilized in this study was established in 1990, the rice plants in the plots without P input showed severe P deficiency symptoms and loss of about 55% of yield, and the plants in the plots without K input exhibited clear K deficient symptoms and lost about 20% of yield. However, how the functional microorganisms such as methanogens and methanotrophs respond to the soil P and K exhausting environments remains unknown. Here we hypothesised that the depleting soil available P and K obviously restricted rice plant growth, and simultaneously, it may also affect the community compositions and functions of methanogens and methanotrophs. And we subsequently used a multi-level approach to resolve the impact of phosphorus and potassium upon the community composition and abundance of both resident (DNA based) and active (mRNA-based) methanogens and methanotrophs and its subsequent influence upon overall methane flux. For clarity, we have changed the title from "Linking phosphorus and potassium deficiency to microbial methane cycling in rice paddies" to "Linking the soils lacking phosphorus & potassium for rice plant to the behaviors of methanogens and methanotrophs and methane emission".

**Comment 3: The relevant reference might be added as follows. Veraart, A. J., Steenbergh, A. K., Ho, A., Kim, S. Y., & Bodelier, P. L. E. (2015). Beyond nitrogen: the importance of phosphorus for CH4 oxidation in soils and sediments. Geoderma, 259-260(December), 337-346.doi:10.1016/j.geoderma.2015.03.025**

We have added this reference in the manuscript.

References

Cai, Z.C., Xing, G.X., Yuan, X.Y., Xu, H., Tsuruta, H, Yagi, K., Minami K.: Methane and nitrous oxide emissions from rice paddy fields as affected by nitrogen fertilisers and water management. Plant Soil, 196(1), 7-14, 1997.

Nishimura, S., Sawamoto, T., Akiyama, H., Sudo, S., Yagi, K.: Methane and nitrous oxide emissions from a paddy field with Japanese conventional water management and fertilizer application. Global Biogeochem. Cy., 18(2), 839-856, 2004.

Towprayoon, S., Smakgahn, K., Poonkaew, S.: Mitigation of methane and nitrous oxide emissions from drained irrigated rice fields. Chemosphere, 59(11), 1547-1556, 2005.

---

## Author Comment (AC2) · 25 Oct 2016

Dear Madam/Sir,

Thank you for your comments and suggestions on our manuscript. We have revised the manuscript accordingly, and highlighted the changes in the revised version. The detailed corrections are listed below point by point:

General comments

The authors performed an interesting study on effects of P and K availability in rice paddy soils on methane emission and methanogen and methanotroph presence and community composition. They find reduced CH4 emission in low P plots, which they attribute to higher methanotroph activity and lower methanogen activity. Effects of

potassium on CH4 emission are less pronounced. Copy numbers of mcrA and pmoA genes are not linked to CH4 emission and hardly show a response to fertilization treatments. Transcripts of pmoA showed differences in the active MOB community between treatments, also dependent on the rice growth stage.

Strong points of the manuscript are the inclusion of mcrA and pmoA transcript analysis in combination with CH4-flux data, and the use of long-term fertilization plots. However, I would like to see improvement on the following issues:

(1) Introduction/Discussion

**Comments: To my understanding, it is quite clear when P or K are limiting plants, but concentrations at which they become limiting to microbes are far less understood. Therefore, I recommend to use terms like 'P/K deficient' with care. For example, your results point at an increase in MOB activity in low P plots, perhaps indicating not so much P-limitation, but an alleviation of excess P? Also, effects may arise from altered (C):N:P:K stoichiometry, rather than concentrations in itself.**

Answer: We agree to the comment that it is less understood about the lack of phosphorus and potassium for plants whether also influence microbial activities. That is the major reason for us to conduct this study. In this study, we observed that the soils without applying P fertilizers for more than 20 years contained very low available P and showed severe limitation to rice growth. But the various influences of low P availability on methanogenic and MOB communities were observed. The population sizes of mcrA-containing methanogens were not obviously affected by the low P availability, but low P concentration exhibited significant restrictions to the transcript abundance, which may in turn limit the activities of methanogens. On the contrary, the copy number of both pmoA gene and gene transcripts were not restricted by low P availability. These phenomena suggested that the low P content restricted plant growth may not limit methanogen and MOB population sizes, but whether it influence the transcription activity of functional gene of methanogens or MOB might rely on functional groups and

environmental factors. Such as the transcription of mcrA gene was blocked by the low P availability while the transcription of pmoA gene was not restricted under same P level. From this study, we can hardly conclude which P level is critical for specific functional microbial group and whether the results is caused by N:P:K stoichiometry, to answer these questions, further investigations will be required. We observed a higher transcription activity of pmoA gene in low P plots based on mRNA analysis, which indicated the plots possess more active MOB and suggested possibly higher activities of methane oxidation. Further studies based on protein level may be useful for assessing the in situ activities of MOB.

**Discuss how, because different MOB respond in different ways, results may strongly depend on the initial community composition. Different soils may react in different ways.**

Answer: We have added related discussion in Line 371 - Line 380 in the manuscript. Details are as follows: "It should be noted that previous studies have documented that soils derived from different parent materials may possess different initial microbial community structure (Ulrich and Becker, 2006; Sheng et al., 2015). Agricultural practices such as crop planting, fertilization and irrigation can modify the initial microbial community structures to some extent based on the initial microbial communities (Fierer et al., 2003). Therefore, different soils may have different methanogenic and methanotrophic community composition structures, their shift patterns in different soils responded to low P availability may vary among different soil types. The variations of the methanogenic and methanotrophic communities in responding to the depleting P and K levels in the paddy soil derived from quaternary red clay may be transferrable to other soils in tendency, but the varying species might be obviously different."

(2) Methods

**Comments: The CH4-flux method is poorly described. Please provide more detail on the method. Where and how were the samples taken? Did the chambers include rice plants? Did you measure time-series? From how many static chambers per plot?**

How many replicate gas samples? How many replicates in time?

Answer: We have added more details about the in situ CH4 flux measurement in Line 125 to Line 136 in the "Materials and methods" section. Additionally, we also added the information about CH4 flux measurement using soil incubation in Line 137 –Line 148. The details are as following: "In situ methane fluxes from the experimental field plots were sampled using static chambers (Shang et al., 2011) at tillering and ripening stages. The sampling chamber was made of PVC with a size of 60×70×90 cm, which was equipped with one circulating fan inside to ensure sufficient gas mixing and wrapped with a layer of sponge to minimize air temperature changes inside the chamber during the period of sampling. After rice transplant, a PVC frame was fixed into a random site in each plot. The top edge of the frame had a groove for filling with water to seal the rim of the chamber. Each frame enclosed 6 rice plants and the height aboveground of the frame is only 5 cm to avoid affecting the growth of rice plants. Gas samples were taken from the chamber headspace with a 30 mL syringe and stored in pre-evacuated vials (Labcolimited high Wycombe UK). At each sampling stage, CH4 fluxes were measured in triplicate plots for all treatments once a day for 3 days. Confirmation of a similar variation trend of CH4 fluxes was observed during these 3 days, we only presented the data from the third day when soil samples were collected in this study. After in situ CH4 flux sampling, fresh soil samples (0–20 cm) were immediately taken from the plots to conduct incubation experiment to determine methane emission rates under controlled environment. The incubation was carried out as follows: after 24 h pre-incubation at 30 oC, equal amounts of fresh soil samples from each treatment (three replicates) were homogenised and 30 g soil (dry weight) was placed into a 250 mL plastic box that can be sealed. For tillering stage samples, soil water content was adjusted to field flooding condition by maintaining 2 cm free surface water. For the ripening stage samples, water content in the soils was adjusted to the same level (50% moisture content, w/w) according to the highest water content of the fresh soil samples. Afterwards, the plastic boxes were sealed and incubated at 30 oC. Headspace gas sampling was conducted at 0 and 60 min, respectively, using a 5 mL syringe and
stored in pre-evacuated vials (Labcolimited high Wycombe UK)."

**Comments: How were total and available P and K determined? Do they reflect availability to plants? / to which extent is P or K unavailable to plants available to microbes?**

Answer: We have added the information about the measurement of soil properties in Line 116 - Line 123 in the "Materials and methods" section. Briefly, total phosphorus (TP) and potassium (TK) were measured by Inductively Coupled Plasma Spectrometry (Agilen, USA) after fusion in NaOH. Available K (AK) was determined by Atomic Absorption Spectroscopy (Seal, Germany) after extraction with NH4OAc. Available P (AP) was measured using UV-Vis Spectrophotometer (PerkinElmer, USA) following extraction with 0.5 M NaHCO3. In this study, we use the variation in plant biomass to show the P and K availability to plants, but to which extent is P or K unavailable to plants available to microbes is a question worthy to be discussed but remains unclear, because the microbial diversity is very high in soil and the requirements for nutrients for different microbes may vary significantly.

(3) Results

**Comments: The figures can be improved. It would be helpful to show hierarchical clustering of the samples based on their T-RFLP profiles, to show which samples/treatments are most similar (per sample class).**

Answer: We have done the cluster analysis and added in the manuscript as Figure 4 (rice tillering stage) and Supplementary Figure 1 (rice ripening stage).

**Comments: Add gene names and DNA vs mRNA copies inside the panels or on the y-axes of the graphs.**

Answer: We have revised in the manuscript.

**Comments: Show correlations between CH4-flux and DNA and mRNA copies, and also present the relation between CH4 flux and mcrA/pmoA transcripts here. You refer**

to these in the discussion but they are missing in the results section.

Answer: We have added the correlation analysis between methanogenic, methanotrophic populations, soil properties and CH4 flux in Line 297 – Line 302 in the "Results" section. The details are as follows: Correlation analysis indicated that the CH4 fluxes from field and soil incubation were significantly correlated to the transcript ratio of mcrA/pmoA (P < 0.05 and P < 0.01, respectively, Table 3) at the tillering stage. In addition, CH4 flux from soil incubation was also significantly correlated with the contents of both total and available phosphorus (TP and AP, P < 0.01), SOC (P < 0.05) and plant biomass (P < 0.05). (4) Discussion ##Comments: Please also better explain why one would expect DNA copy numbers to be less indicative of community functioning than transcripts.

Answer: We have added the sentences "Although other studies also reported that the community structures based on DNA analysis could respond to soil environmental changes and they could reflect the existing state of functional groups (Ahn et al., 2014; Lee et al., 2014; Zheng et al., 2013), the analysis based on gene transcripts are increasingly reported to provide more useful information in understanding the activities of functional microbial communities than the DNA analysis, as gene transcripts are indicative for the active groups against a large resident microbial population (Nicolaisen et al., 2008; Nicol et al., 2008; Freitag et al., 2010)." in Line 331 to Line 357 to explain this question.

**Comments: I am missing some discussion on what these results mean in terms of CH4-mitigation potential? Low emissions seem to come at the expense of plant biomass (and possibly nutritional value?).**

Answer: We have added related discussion in Line 365 – Line 371 in the "Discussion" section. Details are as follows: "In addition, we focused on the analysis of the possible contributions of methanogens and methanotrophs on methane emission in relation to the soil P and K status, but in fact the plant biomass was also affected. Although we

observed that CH4 emission was significantly related to plant biomass, CH4 emissions did not always rely on the plant biomass. For instance, the crop yield was significantly different but the CH4 emission was similar between NPK and –PK treatments. Similar result was also detected by Shang et al. (2011). So, the mitigation of CH4 emission under very low soil P content might be influenced by both poor P nutrition of methanogens and methanotrophs and low plant biomass."

(5) Specific comments

**Comments: Line 31: I would end this sentence at 'transcriptional level', as the relation between 'population size' and DNA copies is debatable.**

Answer: We have deleted the sentence.

**Comments: Line 125, why?**

Answer: We modify the method due to the following reasons: 1) The MP FastPrep can improve the cell lysis efficiency compared to vortex; 2) Using MP FastPrep can save time and labor.

**Comments: Line 254 This seems to conflict with the previous sentence, where members of methylococcus increased. Are these T-RFs representing different species, meaning some methylococcus species increase whereas others decrease?**

Answer: Yes, Methylococcus is a genus we predicted using in silico digestion, unfortunately, we can not specify which species are these T-RFs affiliated to, hence, under P limited condition, maybe some species of this genus were restricted while others were resistant.

**Comments: Line 280 describe how they were influenced by the fertilizer regime**

Answer: We had described the influence of fertilization regime in the following paragraph. Here we focus on the comparative analysis between DNA and mRNA level.

**Comments: Line 286 the 'size' of the resident communities is hardly affected. It**

would be interesting to also discuss the effect of the growth stage of rice on CH4 flux and methanogen and MOB communities.

Answer: We thought that the different size of resident communities between two sampling time might also attribute to the differences in soil water condition, soil nutrient availability and some other factors, so we did not discuss the specific effect of the growth stage of rice on CH4 flux and methanogen and MOB communities in this study.

**Comments: Line 303. How can you be sure that they were P deficient?**

Answer: The significant lower plant biomass in P deficient treatments can verify that the soils were P deficient. Of course, the word "deficiency" here is specific to rice plant rather than soil microbes.

**Comments: Line 325: Add that effects are species specific, different soils may show different effects**

Answer: We have added sentence: "We have observed these effects in our quaternary red soils, but to what extent it is transferrable to other soils remains to be established." in Line 389 – Line 391 in the "Conclusions" section.

**Comments: Technical comments Line 78, round to whole numbers**

Answer: We have revised in the manuscript.

**Comments: Line 87, key nutrients -> phosphorus and potassium**

Answer: We have revised in the manuscript.

**Comments: Line 113, after washing off**

Answer: We have revised in the manuscript.

**Comments: Line 204 different**

Answer: We have revised in the manuscript.

**References**

Nicol, G.W., Leininger, S., Schleper, C., Prosser, J.I.: The influence of soil pH on the diversity, abundance and transcriptional activity of ammonia oxidizing archaea and bacteria. Environ. Microbiol., 10, 2966–2978, 2008. Nicolaisen, M.H., Bælum, J., Jacobsen, C.S., Sørensen, J.: Transcription dynamics of the functional tfdA gene during MCPA herbicide degradation by Cupriavidus necator AEO106 (pRO101) in agricultural soil. Environ. Microbiol., 10, 571–579, 2008.

Sheng, R., Qin H. L., O'Donnell A.G., Huang S., Wu J.S., Wei W.X.: Bacterial succession in paddy soils derived from different parent materials. J. Soils Sediments, 15, 982–992, 2015.

Ulrich., A., Becker, R.: Soil parent material is a key determinant of the bacterial community structure in arable soils. FEMS Microbiol. Ecol., 56, 430–443, 2006.

[Figure]

**Fig. 1.** Revised Fig. 2a

[Figure]

**Fig. 2.** Revised Fig. 2b

[Figure]

[Figure]

**Fig. 3.** Revised Fig. 2c

[Figure]

**Fig. 4.** Revised Fig. 2d

[Figure]

Fig. 5. New added Fig. 4a

[Figure]

[Figure]

[Figure]

(b)

Bray Curtis

**Fig. 6.** New added Fig. 4b

[Figure]

**Fig. 7.** Summplementary Fig. 1a

[Figure]

**Fig. 8.** Summmplementary Fig. 1b

---

## Referee Comment (RC2) · Anonymous Referee #2 · 8 Nov 2016

General comments This is a very interesting study. The authors are trying to elucidate field-scale methane emission flux from microbial ecology perspective, for a better understanding of how anthropogenic activity of fertilizer applications may affect methan-cycling microbes and methane emission in the field. The long-term agricultural field experiment with nutrient deficiency was exploited including –P, -K, and –PK and the balanced fertilization treatments (i.e. NPK). Methane emission fluxes were determined in the field at ripening and tillering stages, the transcriptional activity of key functional genes for methanotrophs (pmoA) and methanogens (mcrA) were determined along with the compostons of these methane-cycling organisms by T-RFLP fingerprinting, plant biomass (above ground and belowground) and soil properties were analyzed. The results showed that that a large amount of CH4 emitted from paddy soil at rice tillering stage (flooding) while CH4 flux was negligible at ripening stage (drying). Compared
to NPK treatment, significantly lower methane flux was observed from P-deficient but not K-deficient fields. Methanotrophic transcript copy number significantly increased in tandem with a decrease in methanogen transcript abundance in P-deficient soils. These results provide important insights on methane-cycling microorganisms in the field thereby contributing to a better understanding of optimization strategy for mitigating methane emission while maintaining crop yield. However, the key message needs to be refined and the focused discussion should be made to establish a correlative link between nutrient constraint and methane emission via plant growth. The major comments are following: (1) Please add a figure showing the correlative relationship between soil phosphorus availability, SOC contents, mRNA and plant biomass and CH4 emission. This would be the key to understand why nutrient-deficiency constrains the growth of rice plant, which may directly or indirectly affect methane-cycling microorganisms, leading to flux variations of methane emission flux in the field. (2) Please convert the plant biomass table as a figure and place it along with methane flux (3) In the text, please discuss the important role of irrigation regime. For example, midseason drainage and the decline of water table at ripening state may lead to significant decline in methane emission flux. Specific comments: (1) L14. It may be more important for plant rather than for the resident microorganisms (2) L15-17. These sentence may be better placed in the text rather than the abstract. (3) L18. It is difficult establish direct link of P and K deficiency to methanogens and methanotrophs. It might be rephrased as plant productivity or crop productivity (4) L20-25. Again, I do not think there is strong evidence in support of conclusion that P deficiency reduced methane emissions via reduced methane production. It may be more appropriate to say that P deficiency constrains the growth of rice plant, leading to lower biomass and methane production. The reason is that the crop biomass may correlate positively with precursors of methanogens. (5) L47. Replace metabolic genes with functional genes (6) L112. Gas sampling means static chamber measurement of CH4 flux in the fied? (7) L116. Methane emission measurement might be merged with soil sampling. static chamber technique can be first described, then soil sampling was conducted in order

to explain the dynamic changes of methane flux in the field. The first section can be the site description only. (8) L121. How were samples kept for transportation before measurement. How to avoid leakage? (9) L124. With slight modification (10) L139. T-RFLP fingerprinting (11) L166. Real-time quantitative PCR (12) The materials and methods can be organized as following. 2.1. site description of long-term field experiment; 2.2. Plant biomass and soil properties; 2.3. Methane emission flux measurement; 2.4. Soil microbial DNA and mRNA extractions; 2.5. Composition and abundance of soil methane-cycling communities (including T-RFLP fingerprinting and Real-time quantitative of soil methane-cycling communities). 2.6. Statistical analysis (13) L202. Please describe the management of rice cultivation. For example, basal fertilizers, top dressing of fertilizers, irrigation regime such as mid-season drainage and so on (14) L235. MOB population size (15) L270-273. The major conclusion of this study here falls short of a reasonable story. For example, the author may come up with few sentence explaining why phosphorus deficiency led to reduction in CH4 emission, while potassium deficiency did not affect net methane emission flux. (16) L298. These different environment conditions should have been in close association with growth status of rice plants under different nutrient regimes. (17) L308-309. Are these organisms methanotrophs, being capable of producing P-liberating enzymes. In addition, if so, it means that there exists the insoluble soil P which can be mineralized by methanotrophs? (18) L309-311. If it is not applied to methanotrophs, please add one or two sentence stating "it should be emphasized that such mechanisms remain unclear in methanotrophs and warrant further study" (19) L323-328. The conclusion should reiterate the key finding of this study. Provide the solid evidence and manage to conclude with a plausible reasoning. For example, the solid evidence is: P deficiency may significantly decrease CH4 flx rate via reducing the activity of methanogens and enhancing the activity of methanotrophs. It may be more appropriate to say that P-deficient soils showed significantly lower CH4 flux. This might be attributed to the reduction of methanogens and the stimulation of methanotrophs that could have adapted to changes in soil physiochemical properties in association with rice plant growth under chronic nutrient constraints.

---

## Author Comment (AC3) · 14 Nov 2016

Dear Madam/Sir,

Thank you for your comments and suggestions on our manuscript.

We have revised the manuscript accordingly, and the detailed corrections are listed below point by point:

General comments

This is a very interesting study. The authors are trying to elucidate field-scale methane emission flux from microbial ecology perspective, for a better understanding of how anthropogenic activity of fertilizer applications may affect methan-cycling microbes and methane emission in the field. The long-term agricultural field experiment with nutrient deficiency was exploited including –P, -K, and –PK and the balanced fertilization treatments (i.e. NPK). Methane emission fluxes were determined in the field at ripening and tillering stages, the transcriptional activity of key functional genes for methanotrophs (pmoA) and methanogens (mcrA) were determined along with the compostions of these methane-cycling organisms by T-RFLP fingerprinting, plant biomass (above ground and belowground) and soil properties were analyzed. The results showed that that a large amount of $CH_4$ emitted from paddy soil at rice tillering stage (flooding) while $CH_4$ flux was negligible at ripening stage (drying). Compared to NPK treatment, significantly lower methane flux was observed from P-deficient but not K-deficient fields. Methanotrophic transcript copy number significantly increased in tandem with a decrease in methanogen transcript abundance in P-deficient soils. These results provide important insights on methane-cycling microorganisms in the field thereby contributing to a better understanding of optimization strategy for mitigating methane emission while maintaining crop yield.

However, the key message needs to be refined and the focused discussion should be made to establish a correlative link between nutrient constraint and methane emission via plant growth. The major comments are following:

**Comment (1): Please add a figure showing the correlative relationship between soil phosphorus availability, SOC contents, mRNA and plant biomass and $CH_4$ emission. This would be the key to understand why nutrient-deficiency constrains the growth of rice plant, which may directly or indirectly affect methane-cycling microorganisms, leading to flux variations of methane emission flux in the field.**

Answer: We have added the correlation analysis between methanogenic, methanotrophic populations, soil properties and $CH_4$ flux as requested in lines 215-224, line 304-312 and Fig. 5. The details are as below:

Line 215-224: Soil properties such as pH, soil organic carbon and total nitrogen together with gene abundance between the treatments were compared by ANOVA analysis using the Statistical Package for Social Science (SPSS 13, SPSS Inc., Chicago, IL, USA). Significance among means was identified using least significant differences. Pearson correlation analysis between CH4 flux, soil properties, plant biomass and population size of resident and active methanogens and methanotrophs was also performed using SPSS. Redundancy analysis (RDA) was used to characterize the relationship between soil properties, plant biomass and the community structures of methanogens and methanotrophs using CANOCO statistical package for Windows 4.5 (Biometris, Wageningen, Netherlands). A Mantel test based on 499 random permutations was used to examine the significant correlations between the differences in soil properties plant biomass and microbial communities.

Line 304-312: Correlation analysis indicated that the CH4 fluxes from field and soil incubation were significantly correlated to the transcript ratio of mcrA/pmoA (P < 0.05 and P < 0.01, respectively, Table 3) at the tillering stage. In addition, CH4 flux from soil incubation was also significantly correlated with the contents of both total and available phosphorus (TP and AP, P < 0.01), SOC (P < 0.05) and plant biomass (P < 0.05). Redundancy analysis (RDA) indicated that P-deficient induced changes in soil physiochemical properties, such as SOC, TP, AP contents in tandem with plant biomass, were important factors driving community structure shifts of active (mRNA based) methanogens and methanotrophs (Fig. 5).

**Comment (2): Please convert the plant biomass table as a figure and place it along with methane flux**

Answer: We have convert plant biomass table and CH4 flux as Fig. 1

**Comment (3): In the text, please discuss the important role of irrigation regime. For example, midseason drainage and the decline of water table at ripening state may lead to significant decline in methane emission flux.**

Answer: We have added discussion in line 382-388:

"Besides, soil water management has been widely known to play an important role in regulating CH4 emission (Cai et al., 1997; Nishimura et al., 2004; Towprayoon et al., 2005). We observed that the CH4 flux were much lower at rice ripening stage when soil was drying than that at tillering stage when soil was under flooding. These phenomena were also reported by previous studies that showing midseason drainage and the disappearance of the water layer induced significant decline in methane emission flux, which might associated with the reduction in methane production and increase in the oxidation of CH4 under drying soil environment (Nishimura et al., 2004; Towprayoon et al., 2005)."

Specific comments:

**Comment (1): L14. It may be more important for plant rather than for the resident microorganisms**

Answer: It has been rewrite as below:

"Nutrient status in soil is crucial for the growth and development of plant and resident soil microorganisms. Soil methanogens and methanotrophs can be affected by soil nutrient availabilities and plant growth, which in turn modulate methane (CH4) emissions."

**Comment (2): L15-17. These sentence may be better placed in the text rather than the abstract.**

Answer: We have deleted this sentence.

**Comment (3): L18. It is difficult establish direct link of P and K deficiency to methanogens and methanotrophs. It might be rephrased as plant productivity or crop productivity**

Answer: It has been rewrite as below:

"Nutrient deficient has been shown to constrain plant growth, however, whether nutrient limitation for plant can also influence the methanogenic and methanotrophic

communities and their functions are remained unclear. Here, we assessed whether deficits in soil available phosphorus (P) and potassium (K) modulated the activities of methanogens, methanotrophs in a long term (20 y) experimental system undergoing limitation in either one or both nutrients."

**Comment (4): L20-25. Again, I do not think there is strong evidence in support of conclusion that P deficiency reduced methane emissions via reduced methane production. It may be more appropriate to say that P deficiency constrains the growth of rice plant, leading to lower biomass and methane production. The reason is that the crop biomass may correlate positively with precursors of methanogens.**

Answer: It has been rewrite as below:

"Results showed that a large amount of CH4 emitted from paddy soil at rice tillering stage (flooding) while CH4 flux was minimum at ripening stage (drying). Compared to NPK treatment, the soils without P input significantly reduced methane flux rates, whereas without K input did not. Under P limitation, methanotroph transcript copy number significantly increased in tandem with a decrease in methanogen transcript abundance, suggesting that P-deficient induced changes in soil physiochemical properties in tandem with rice plant growth might constrain the activity of methanogens, whereas the methanotrophs might be adaptive to this soil environment. In contrast, lower transcript abundance of both methanogen and methanotrophs were observed in K-deficient soils. Assessments of community structures based upon transcript indicated that soils deficits in P induced greater shifts in the active methanotrophic community than K-deficient soils while similar community structures of active methanogens were observed in both treatments."

**Comment (5): L47. Replace metabolic genes with functional genes**

Answer: We have revised as requested.

**Comment (6): L112. Gas sampling means static chamber measurement of CH4 flux**

in the fied?

Answer: Yes, we also supplement some details of the measurement of CH4 flux rate using soil incubation experiment in line 128-142 in the revised version.

**Comment (7): L116. Methane emission measurement might be merged with soil sampling. static chamber technique can be first described, then soil sampling was conducted in order to explain the dynamic changes of methane flux in the field. The first section can be the site description only.**

Answer: We have revised as requested.

**Comment (8): L121. How were samples kept for transportation before measurement. How to avoid leakage?**

Answer: Gas samples were stored in pre-evacuated vials (Labcolimited high Wycombe UK), which can prevent gas leakage.

**Comment (9): L124. With slight modification**

Answer: We have revised as required.

**Comment (10): L139. T-RFLP fingerprinting**

Answer: We have revised as required.

**Comment (11): L166. Real-time quantitative PCR**

Answer: We have revised as required.

**Comment (12): The materials and methods can be organized as following. 2.1. site description of long-term field experiment; 2.2. Plant biomass and soil properties; 2.3. Methane emission flux measurement; 2.4. Soil microbial DNA and mRNA extractions; 2.5. Composition and abundance of soil methane-cycling communities (including T-RFLP fingerprinting and Real-time quantitative of soil methane-cycling communities). 2.6. Statistical analysis**

Answer: We have revised as required.

**Comment (13): L202. Please describe the management of rice cultivation. For example, basal fertilizers, top dressing of fertilizers, irrigation regime such as mid-season drainage and so on**

Answer: We have added some information regarding the management of rice cultivation, including fertilization and water management in line 105- 112, the details are as below: "For the late rice-cropping season when we sampling, urea was applied with three splits, 40% as basal fertilizer, 50% as tillering fertilizer and 10% as panicle fertilizer. The P and K fertilizers were applied as basal fertilizers before rice transplanting. The basal fertilizers were well incorporated into the soil by plowing to 10-20 cm depth 2 days before rice planting, and the top-dressing was surface broadcasted. Consistent with the water management in local late rice-cropping system, flooding was initiated after early rice harvest before late rice transplanting, and maintained until 10 days before rice harvesting. During this period, a 7 days drainage episode was implemented at late tillering stage."

**Comment (14): L235. MOB population size**

Answer: In this section, we presented both size and community structure of MOB population, hence we use MOB population here.

**Comment (15): L270-273. The major conclusion of this study here falls short of a reasonable story. For example, the author may come up with few sentence explaining why phosphorus deficiency led to reduction in CH4 emission, while potassium deficiency did not affect net methane emission flux.**

Answer: We describe this in line 323-381.

**Comment (16): L298. These different environment conditions should have been in close association with growth status of rice plants under different nutrient regimes.**

Answer: We have deleted this sentence.

**Comment (17): L308-309. Are these organisms methanotrophs, being capable of producing P-liberating enzymes. In addition, if so, it means that there exists the insoluble soil P which can be mineralized by methanotrophs?**

Answer: We do not have solid evidence to verify that methanotrophs can produce P-liberating enzyme now, here we just speculated that some methanotrophs may produce P-liberating enzymes based on the studies carried on other microbes, further studies based on methanotrophic strains still need to be investigated.

**Comment (18): L309-311. If it is not applied to methanotrophs, please add one or two sentence stating "it should be emphasized that such mechanisms remain unclear in methanotrophs and warrant further study"**

Answer: We have added "Although we do not know the real mechanisms about the enrichment of Methylococcus/Methylocaldum under such a poor soil P nutritional status, it could be speculated that the possible adaptations of these MOB groups to a P deficient environment might be attributed to one or more adaptive strategies." in line 363-366.

**Comment (19): L323-328. The conclusion should reiterate the key finding of this study. Provide the solid evidence and manage to conclude with a plausible reasoning. For example, the solid evidence is: P deficiency may significantly decrease CH4 flx rate via reducing the activity of methanogens and enhancing the activity of methanotrophs. It may be more appropriate to say that P-deficient soils showed significantly lower CH4 flux. This might be attributed to the reduction of methanogens and the stimulation of methanotrophs that could have adapted to changes in soil physiochemical properties in association with rice plant growth under chronic nutrient constraints.**

Answer: Conclusion now appears as follows:

P-deficient soils showed significantly lower CH4 flux. This might be attributed to the restriction of methanogens and the stimulation of methanotrophs that could have

adapted to changes in soil physiochemical properties in association with rice plant growth under chronic nutrient constraints. In contrast, K-deficient did not affect the CH4 flux, which might be caused by the reductions of both methanogenic and methanotrophic activities. Comparatively, more variations within community composition of the active methanotrophs were observed in P-deficient soils than that in K-deficient soils, whereas both P- and K-deficient soils shared similar active methanogenic community structures. We have observed these effects in our quaternary red soils, but to what extent it is transferrable to other soils remains to be established.

Please also note the supplement to this comment:
http://www.biogeosciences-discuss.net/bg-2016-213/bg-2016-213-AC3-supplement.pdf

**Supplement:**

**Linking the soils lacking phosphorus & potassium for rice plant to the behaviors of methanogens and methanotrophs and methane emission**

**Rong Sheng[1,3], Anlei Chen[1], Miaomiao Zhang[1], Andrew S Whiteley[2,3], Deepak Kumaresan[2,3], Wenxue Wei[1,3]**

[1]Key laboratory of Agro-ecological Processes in Subtropical Regions and Taoyuan Agro-ecosystem Research Station, Soil Molecular Ecology Section, Institute of Subtropical Agriculture, Chinese Academy of Sciences, Changsha 410125, China

[2]School of Earth and Environment, The University of Western Australia, Perth 6009, Western Australia

[3]ISA-CAS and UWA Joint Laboratory for Soil Systems Biology, Institute of Subtropical Agriculture, Chinese Academy of Sciences, Changsha 410125, China

*Correspondence to*: Wenxue Wei (wenxuewei@isa.ac.cn)

**Abstract** Nutrient status in soil is crucial for the growth and development of plant and resident soil microorganisms. Soil methanogens and methanotrophs can be affected by soil nutrient availabilities and plant growth, which in turn modulate methane ($CH_4$) emissions. Nutrient deficient has been shown to constrain plant growth, however, whether nutrient limitation for plant can also influence the methanogenic and methanotrophic communities and their functions are remained unclear. 
[revised manuscript text omitted]

---

## Author Response (AR1)

**Response to reviewers' comments**

Dear Editor,

Thank you for your comments and suggestions on our manuscript.

5    We have revised the manuscript accordingly, and highlighted the changes in the revised version. The detailed corrections are listed below point by point:

The major criticisms have been addressed and few minor concerns are the following:

(1) L1-2. The title may be rephrased as follows: Transcriptional activities of methanogens and methanotrophs vary with methane emission flux in rice soils under chronic nutrients constraints of

10    phosphorus and potassium; or Field methane emission is associated with transcriptional activities of methanogens and methanotrophs in rice soils under chronic nutrients constraints of phosphorus and potassium.

Answer:    We have changed the title to "Transcriptional activities of methanogens and methanotrophs vary with methane emission flux in rice soils under chronic nutrients constraints of phosphorus and

15    potassium"

(2) L13. It may be rephrased: Nutrient status in soil is crucial for the growth and development of plants which indirectly/directly affect the ecophysiological functions of resident soil microorganisms.

Answer:    We have revised as required.

(3) L17. Delete are.

20    Answer: We have deleted this sentence.

(4) L30. A concluding remark might be added. For example, these results suggest that population dynamics of methanogens and methanotrophs vary along with plant growth stage and soil property changes under nutrient deficiency.

Answer: We have added a sentence "These results suggested that the population dynamics and functions

25    of methanogens and methanotrophs could vary along with the changes in plant growth states and soil properties induced by nutrient deficiency."

(5) L73. Pls add a reference about nitrogen fertilizers on methane emission: Bodelier PLE, Roslev P, Henckel T, Frenzel P (2000). Stimulation by ammonium-based fertilizers of methane oxidation in soil around rice roots. Nature 403: 421-424

30    Answer: Have added.

(6) L409. It may be rephrased as following: but to what it can be generalized remains unclear, considering the remkarble soil heterogenity.

Answer: It has been revised as "We have observed these effects in our quaternary red soils, but to what extent it can be generalized remains unclear, considering the remarkable soil heterogeneity."

**Transcriptional activities of methanogens and methanotrophs vary with methane emission flux in rice soils under chronic nutrients constraints of phosphorus and potassium**Linking the soils lacking phosphorus & potassium for rice plant to the behaviors of methanogens and methanotrophs and methane emission

**Rong Sheng[1,3], Anlei Chen[1], Miaomiao Zhang[1], Andrew S Whiteley[2,3], Deepak Kumaresan[2,3], Wenxue Wei[1,3]**

[1]Key laboratory of Agro-ecological Processes in Subtropical Regions and Taoyuan Agro-ecosystem Research Station, Soil Molecular Ecology Section, Institute of Subtropical Agriculture, Chinese Academy of Sciences, Changsha 410125, China

[2]School of Earth and Environment, The University of Western Australia, Perth 6009, Western Australia

[3]ISA-CAS and UWA Joint Laboratory for Soil Systems Biology, Institute of Subtropical Agriculture, Chinese Academy of Sciences, Changsha 410125, China

*Correspondence to*: Wenxue Wei (wenxuewei@isa.ac.cn)

**Abstract** Nutrient status in soil is crucial for the growth and development of plants which indirectly/directly affect the ecophysiological functions of and resident soil microorganisms. Soil methanogens and methanotrophs can be affected by soil nutrient availabilities and plant growth, which in turn modulate methane ($CH_4$) emissions. Nutrient deficient has been shown to constrain plant growth,

however, whether nutrient limitation for plant can also influence the methanogenic and methanotrophic communities and their functions are remained unclear. 
[revised manuscript text omitted]

645

---

## Author Response (AR2)

**Response to editor's comments**

Dear Editor,

Thank you for your comments and suggestions on our manuscript.

We have revised the manuscript accordingly, and highlighted the changes in the revised version. The detailed corrections are listed below point by point:

The ms fits well with the scope of BG, and I would like to have your attention for a minor concern before publication

(1) L211. methane-cycling microbes might be replaced by methanotrophs and methaogens

Answer:    Have revised

(2) the subtitles in L283 and L311. please use the term consistently. (under depleting soil available P and K circumstance; under exhausting soil available P and K circumstance). I guess it may not be depleted for P and K?. maybe the subtitle can be used with "Shifts of methanogenic populations and transcripts in soil with P and K deficiency.

Answer:    The subtitles have been changed to "Shifts of methanogenic populations and transcripts in soils with P and K deficiency" and "Shifts of methanotrophic populations and transcripts in soils with P and K deficiency"

[revised manuscript text omitted]